# Neutrophils Promote Glioblastoma Tumor Cell Migration after Biopsy

**DOI:** 10.3390/cells11142196

**Published:** 2022-07-14

**Authors:** Na Chen, Maria Alieva, Tom van der Most, Joelle A. Z. Klazen, Arabel Vollmann-Zwerenz, Peter Hau, Nienke Vrisekoop

**Affiliations:** 1Department of Respiratory Medicine, Center of Translational Immunology, University Medical Center Utrecht, Lundlaan 6, Mail room KC02.085.2, 3584 EA Utrecht, The Netherlands; jilindaxuechenna@163.com (N.C.); tom.vandermost@radboudumc.nl (T.v.d.M.); j.a.z.klazen-2@umcutrecht.nl (J.A.Z.K.); 2Princess Máxima Center for Pediatric Oncology, Heidelberglaan 25, 3584 CS Utrecht, The Netherlands; m.alieva@prinsesmaximacentrum.nl; 3Oncode Institute, 3521 AL Utrecht, The Netherlands; 4Wilhelm Sander NeuroOncology-Unit and Department of Neurology, University Hospital Regensburg, 93053 Regensburg, Germany; arabel.vollmann@klinik.uni-regensburg.de (A.V.-Z.); peter.hau@klinik.uni-regensburg.de (P.H.)

**Keywords:** neutrophils, glioblastoma, tumor cell migration

## Abstract

Glioblastoma is diagnosed by biopsy or, if clinically feasible, tumor resection. However, emerging evidence suggests that this surgical intervention may increase the risk of tumor cell spread. It has been hypothesized that the damage to the tumor leads to infiltration of immune cells that consequently form an environment that favors tumor cell motility. In mouse glioma models, it was previously found that biopsy induced migration of tumor cells in vivo and that recruitment of monocytes from the blood was involved in this effect. However, the role of neutrophils in this process is still unclear. Here, we study the contribution of neutrophils on the pro-migratory effect of surgical interventions in glioma. Using repetitive intravital microscopy, in vivo migration of glioma tumor cells before and after biopsy was compared in mice systemically depleted of neutrophils. Interestingly, macrophages/microglia were almost completely absent from neutrophil-depleted tumors, indicating that neutrophils may be indirectly involved in biopsy-induced migration of glioma tumor cells through the recruitment of macrophages to the tumor. To further investigate whether neutrophils have the potential to also directly promote glioblastoma tumor cell migration, we performed in vitro migration assays using human neutrophils. Indeed, wound-healing of human primary glioblastoma tumor cell lines was promoted by human neutrophils. The pro-migratory effects of human neutrophils on glioblastoma tumor cells could also be recapitulated in transwell migration assays, indicating that soluble factor(s) are involved. We therefore provide evidence for both an indirect and direct involvement of neutrophils in tumor spread following biopsy of glioblastoma tumors.

## 1. Introduction

Glioblastoma, or grade IV/4 glioma, is the most prevalent and aggressive brain tumor in adults [1,2]. Median survival of patients is around 16 months and both tumor growth and local dissemination have been found to be associated with decreased survival [3,4]. Gliomas are biopsied or resected before combined radio-chemotherapy. Although these surgical procedures are prognostically beneficial, they have been associated with accelerated tumor growth and the formation of metastases in other solid tumors [5,6]. Experimental studies in mice bearing a range of solid tumors similarly found that tumor biopsy or complete resection can lead to an increased number of metastases or to the promotion of proliferation and invasion of the primary tumor [7,8,9]. It was hypothesized that the damage to the tumor and surrounding tissue triggers an inflammatory reaction that creates a pro-metastatic environment [8,9,10]. Several studies have investigated how biopsy affects the immune microenvironment in different tumor types, finding increased numbers of (‘alternatively activated’) macrophages [9,11,12], regulatory T-cells [10], neutrophils [8,9], and eosinophils [12] at the biopsy site or around the tumor. In interaction with tumors, macrophages have been thought to either adopt a pro-tumoral phenotype (alternatively activated (or M2) macrophages) or anti-tumoral phenotype (M1 macrophages), although the strict duality has recently been questioned [13]. The potential for alternatively activated macrophages to create an immunosuppressive environment promoting cancer disease progression has been well described [13]. An increase in the number of (alternatively activated) macrophages is therefore thought to mediate the pro-metastatic response following biopsy [9,11].

Neutrophils have received less attention in the past, as they were deemed too short-lived to affect a chronic disease such as cancer, and also because their immunosuppressive role has only recently been discovered [14]. Recent developments, however, show that there is profound interaction between tumor cells and neutrophils. Tumors produce factors such as G-CSF and GM-CSF to promote the release of neutrophils from the bone marrow and secrete pro-survival factors to increase their life-span [14,15]. For several tumor types, a high neutrophil-to-lymphocyte ratio (NLR) or high number of intra-tumoral neutrophils is associated with poor prognosis in patients and is used as a prognostic marker [16,17,18,19,20,21]. It has been postulated that similar to macrophages, neutrophils may be induced to have a pro-tumoral (N2 neutrophils) or anti-tumoral (N1 neutrophils) phenotype, depending on factors released by the tumor [22]. Furthermore, in inflammatory conditions including cancer, myeloid cells are found with immunosuppressive capabilities, called myeloid-derived suppressor cells (MDSC). The granulocytic subset (G-MDSC) can be considered to be neutrophils with immunosuppressive capabilities [14,23]. Accordingly, in a mouse model for breast cancer, infiltration of MDSCs in the tumor following biopsy creates an immunosuppressive environment favoring metastatic spreading of the tumor to the lung [8].

Recently, we investigated the effect of needle biopsy on progression of glioblastoma tumors. Alieva et al. show in patients with multifocal glioblastoma that biopsied tumor sites increase more in volume compared to non-biopsied sites. Additionally, we show that biopsy-like injury promotes local tumor cell invasion and division in a mouse model for glioblastoma [9]. Both macrophages and neutrophils are recruited to the glioblastoma upon injury. The exact mechanism for these observations has not been elucidated so far. However, depleting monocytes using clodronate liposomes prevents the increased in vivo migration of glioblastoma cells upon biopsy, indicating the involvement of these cells [9]. CCL-2 is a chemokine secreted by tumor and stromal cells and mediates recruitment of monocytes and neutrophils, both expressing the receptor CCR2 [9,24,25]. CCL-2 blockade reduces the percentage of migratory glioblastoma cells upon biopsy below that observed for control and clodronate-liposome-treated mice [9].

Our own observations and the increased effectiveness of CCL-2 blockade prompted us to further investigate the role of neutrophils in glioma tumor cell migration.

## 2. Materials & Methods

### 2.1. Tumor Cell Lines

The H2b-Dendra2 fluorescent mouse glioma cell line GL261 was generated using a standard lentiviral transduction protocol with a pLV CMV-H2B-Dendra2 vector [9]. GL261 H2b-dendra2 mouse glioma cells were cultured in Dulbecco’s Modified Eagle’s Medium + GlutaMAX (DMEM; GIBCO, Invitrogen Life Technologies, Paisley, UK) supplemented with 10% (*v*/*v*) fetal bovine serum (Sigma, St. Louis, MO, USA), 100 μg/mL streptomycin, and 100 U/mL penicillin (Invitrogen Life Technologies, Paisley, UK). Green fluorescent human glioblastoma BTIC cell lines (BTIC10 and BTIC13) were lentivirally transduced with a U57 pHR SFFV GFP plasmid [26,27,28]. BTIC cell-lines were cultured in RHB-A stem-cell permissive medium (Stem Cell Sciences) supplemented with 20 ng/mL human recombinant EGF/basic FGF (Peprotech). The cell-lines were cultured at 37 °C with 5% CO_2_ and grew in adhesive monolayers. Cells were disassociated from culture plates using 1:1 Accutase/PBS for passaging.

### 2.2. Mice

All animal experiments were carried out in accordance with the guidelines of the Animal Welfare Committee of the Royal Netherlands Academy of Arts and Sciences, the Netherlands. The experimental protocols used in this manuscript were approved by the Centrale Commissie Dierproeven (CCD, approval code AVD801002015125, approval date 22 November 2016). C57BL/6 mice (age 8–12 weeks) were used for the experiments. Mice were housed at the animal facility in the Hubrecht Institute and received food and water ad libitum. Implantation of tumor cells, cranial imaging window (CIW) placement, and biopsy-like injury have been described extensively elsewhere [6,9]. In short, on day 0, mice were sedated, the head was shaved, and an incision was made on the top part of the head. A circular hole was drilled over the right parietal bone and 1 × 10^5^ GL261 H2b-dendra2 glioma cells, suspended in 3 µL PBS, were injected at a depth of 0.5 mm. Subsequently, a chronic CIW was placed. Three mice received intraperitoneal injection of 100 µg Ly6G antibodies (1A8; BioXcell, Lebanon, PA, USA) on days 8 and 11. On day 12, the CIW was replaced and mice received biopsy-like injury. The injury consisted of four needle punctures with a 25G needle in the tumor at a depth of 1 mm. The needle contained fluorescent beads allowing the identification of the biopsy area. On day 13, the mice were sacrificed. Blood was obtained via heart puncture and the tumor was harvested. The experiments were performed in two independent groups. These procedures were performed by the same researcher in the same time-period as control data without Ly6G antibody depletion with biopsy (six mice) and without biopsy (six mice), as previously reported by Alieva et al. [9].

### 2.3. Mouse Intravital Imaging & In Vivo Cell Tracking

Mice were intravitally imaged through the CIW on days 11 and 13 with a Leica TCS SP5 AOBS two-photon microscope as described before *(*Figure 1 [9]. Mice were imaged for a duration of 2 h and an image was taken every 20 min. A z-stack was made of different positions of the tumor at a maximal depth of 300 µm, depending on the position of the tumor, with a 3 µm interval between the z-slices. Z-drifts were corrected for using Jittering Corrector software. Three z-slices were merged to create a maximum projection of 6 µm thickness. Maximum projections were corrected for deformations (e.g., caused by breathing of the animal) using the Match software. For analysis, three random positions were chosen and every other maximum projection was analyzed for the complete time-period using the ‘MtrackJ’ plug-in for ImageJ. Per maximum projection, 2–50 cells were tracked and the displacement was determined, adding up to a total number of 1081–3562 tracks per animal. Vector speed is the displacement/hour. Using the cut-off value of 4 µm/h vector speed to define migratory cells, the percentage of migratory cells before and after biopsy was determined. The fold change is the %migratory cells after/before biopsy. Data of the fold change in migratory cells from biopsy mice without neutrophil depletion were reused from Alieva et al. [9].

### 2.4. Immunostaining of Mouse Brain Slices

Tumor cryosections of 14µm thickness were stained overnight with biotin-conjugated F4/80 (mf48015; Invitrogen). Secondary staining was performed with fluorophore-conjugated streptavidin (S-21374; Life Technologies). A Leica SPE confocal microscope was used to make tile scans, and the LaSAF software was used to quantify the cells. The biopsy area could be identified based on the presence of fluorescent beads injected during biopsy-like injury. The number of F4/80^+^ cells was determined in both the biopsy and non-biopsied area and normalized for the size of the area (mm^2^). The macrophage numbers from biopsy mice without neutrophil depletion were reused from Alieva et al. [9].

### 2.5. Flow Cytometry of Mouse Blood Cells

Blood was diluted with 1 mL red blood cell lysis buffer (NH_4_Cl) per 100 µL blood and kept on ice for 10 min. Cells were spun down (5 min at 4 °C 500 RCF), resuspended in 1 mL NH_4_Cl and kept on ice for another 3 min. Afterwards, 3 mL of FACS buffer (PBS + 5% FCS + 5 mM EDTA) was added and cells were spun down again (4 min at 4 °C 500 RCF). The pellet was resuspended in 500 µL FACS buffer for flow cytometric analysis on a FACS Calibur. The neutrophil percentages in blood from biopsy mice without neutrophil depletion were reused from Alieva et al. [9].

### 2.6. Human Blood and Neutrophil Isolation

Human blood samples were collected by venapuncture from anonymous, male and female, healthy volunteers between the age of 18–65 years. All donors gave informed consent under protocols approved by the Biobanks Review committee of the University Medical Center Utrecht (approval code 18/774, approval date 25 June 2013). Blood with sodium heparin anticoagulant was diluted 1:1 with PBS^++^ containing 40 g/L albumin and 3.2% sodium citrate and separated by Paque-Ficoll (GE Healthcare) density centrifugation for 20 min at RT 2000 rpm. After removal of the plasma, PBMCs, and most of the Ficoll, remaining cells were suspended in red blood cell lysis buffer for 10–15 min on ice. Cells were spun down (1500 rpm 5 min 4 °C) and washed with red blood cell lysis buffer and subsequently with PBS^++^. Cells were kept in culture medium on ice until further use.

### 2.7. Wound-Healing Assay

For the wound-healing assay, a minimum of six different donors in three or more different experimental groups were included. Silicone culture inserts (IBIDI) were used, in which tumor cells were grown in two separate compartments, leaving a cell-free gap between the tumor cells after removal of the insert. Tumor cells were seeded in both compartments of the culture insert on a culture-treated 24-well plate in RHB-A medium until confluency was reached. Subsequently, culture medium was aspirated and the culture insert removed. Tumor cells were resuspended with culture medium including 6 × 10^4^ human neutrophils or culture medium only. The Incucyte live cell imaging system was used to make hourly images and allowed the tumor cells to be left unperturbed in the incubator throughout the entire period. In order to quantify migration into the cell-free area, cells were masked on ImageJ, transforming the images into binary images in which cell-covered pixels are given the value 255 and background 0. By defining the cell-free area, migration of cells into the gap is represented by increases in the average pixel intensity in the cell-free area. Cells were judged confluent at the end of imaging and percentage confluency over time was determined by comparing the average pixel intensity in the present time point to that at the end of imaging.

### 2.8. Transwell Migration Assay

For the transwell migration assay, a minimum of six different donors in three or more different experimental groups were included. The tumor cells were seeded in a transwell with a pore size of 8 µm (Corning) that were placed on top of wells of normal tissue culture plates. In the transwell, 1 × 10^5^ tumor cells were seeded in 300 µL medium and were allowed to migrate through 8µm pores on the bottom of the transwell. On the bottom of the well, 700 µL culture medium including 5 × 10^5^ human neutrophils or culture medium only was pipetted, ensuring that the culture medium extended above the bottom of the transwell. Using this set-up, the only contact between the neutrophils and the tumor cells is via the culture medium, preventing contact-dependent pro-migratory effects of neutrophils. After 24 h, the transwells were carefully removed and images were taken of the bottom of the plates using the Incucyte. The number of fluorescent tumor cells was determined by applying a mask on fluorescence and automatically counting the particles on ImageJ. To determine the average number of transmigrated cells, images of five positions within the same well were analyzed.

### 2.9. Statistics

Results are shown as median with range (min-max). When > 2 medians were compared, Kruskal–Wallis test was used, and when two medians were compared, the Mann–Whitney U-test was used. For paired comparison, the two-tailed Wilcoxon matched-pairs signed rank test was used, * *p* < 0.05, ** *p* < 0.01.

## 3. Results

### 3.1. In Vivo Biopsy Induces a Neutrophil-Dependent Increase in Motility of Mouse Glioma Tumor Cells

We first evaluated the role of neutrophils on glioblastoma tumor cell migration in an in vivo mouse biopsy–injury model published earlier [9]. Brain surgery was performed to inject GL261 mouse glioma cells expressing a nuclear fluorescent protein. Consecutively, a chronic cranial imaging window was implanted that allowed tracking of the movement of glioma tumor cells in vivo at later time points [6,9]. Within one week, the mice developed an invasive brain tumor, in line with the aggressive nature of glioblastoma tumors in humans [1,9]. In this model, biopsy induced glioblastoma tumor cell motility, coinciding with a 3.3-fold increase in the number of neutrophils and a 2-fold increase in the number of macrophages [9]. In order to study the role of neutrophils, mice were depleted for neutrophils by two injections with Ly6G-antibodies. The effect of biopsy on in vivo tumor cell migration was determined by repeated intravital microscopy on the day before (day 11) and after biopsy-like injury (day 13) (Figure 1). Using this approach, individual GL261 glioma cell movement could accurately be tracked in a horizontal plane at multiple depths within the tumor mass; a migration speed of 1081–3562 tracks per mouse was calculated. The same cut-off speed of 4 μm/h as previously used to define migratory cells was applied to determine the percentage of migratory tumor cells both before and after biopsy for three Ly6G antibody-treated mice.

As published previously [9], the fold change of migratory tumor cells at day 13 over day 11 decreased if no biopsy was induced. In contrast, the fold change of migratory tumor cells significantly increased after biopsy (Figure 2A,B). In neutrophil-depleted mice, the fold change in migratory cells upon biopsy significantly decreased compared to controls without neutrophil depletion (Figure 2A). Furthermore, the speed of migratory cells was significantly decreased in neutrophil-depleted tumors after biopsy compared to control biopsied mice without neutrophil depletion (Figure 2B and Appendix A). Of note, we have previously reported that a control IgG did not change GL261 migration or cell division [9].

Directly after the intravital imaging session post-biopsy, the mice were sacrificed and blood was analyzed by flow cytometry to evaluate depletion efficacy. Since remaining neutrophils might be saturated with Ly6G depletion mAbs, we could not use this antibody to detect neutrophils in blood and tumor tissue. To avoid overestimating the depletion effect, residual neutrophils were identified based on forward scatter (FSC) and side scatter (SSC), and the percentage of residual neutrophils in the blood for the three mice was 3.95%, 5.42%, and 9.38% (of all FSC/SSC gated events; Figure 2C). The median percentage of neutrophils in control tumor bearing mice was 18% of total cells *(*Figure 2C). Thus, neutrophil numbers in the blood were reduced but not completely depleted following Ly6G antibody treatment, as has been described before [29,30].

In order to observe the effect of Ly6G-depletion on the number of macrophages in the tumor, brain cryosections were stained with F4/80 to quantify the number of macrophages in biopsied and non-biopsied tumor tissue for each mouse. The biopsied area was identified by the presence of fluorescent beads that were injected during biopsy-like injury (Appendix A). When comparing the numbers of macrophages in the brain of the Ly6G antibody-treated mice to the untreated biopsied mice, macrophages were significantly reduced in both biopsied and non-biopsied tumor tissue (Figure 2D).

### 3.2. Neutrophils Promote In Vitro Wound-Closure of Human Glioblastoma Tumor Cells

As in vivo neutrophil depletion results in the local depletion of macrophages from the tumor, our mouse model was unsuited for investigating independent effects of neutrophils. In order to study the direct effect of neutrophils on tumor cell motility, the effect of human neutrophils on human tumor cell line motility was determined in vitro. Human glioblastoma tumor cell lines were derived from high-grade glioma patients. The cell lines were enriched for brain-tumor-initiating cells (BTICs), as recurring malignant gliomas are believed to be derived from a population of cells with stem-cell-like characteristics [31]. These cell-lines (BTIC10 and BTIC13) have been described extensively and used to investigate human glioblastoma tumor cell behavior in vitro [26,27,28]. In order to quantify the migratory behavior of BTIC, a tumor-cell-free wound area was created. The time until confluency was reached in the wound area was taken as a measure of migratory capacity. In the presence of neutrophils, wound closure time was significantly reduced for both BTIC10 and BTIC13 glioblastoma tumor cell lines. Neutrophils did not promote BTIC tumor cell proliferation (Appendix A), indicating that the faster wound closure time in the presence of neutrophils reflected an increase in migration of human glioblastoma cells (Figure 3).

### 3.3. Soluble Factor(s) from Neutrophils Increase Transmigration of Human Glioblastoma Cells

To establish whether the increase in tumor cell migration by neutrophils was cell-contact dependent, we performed transwell migration assays with the human glioblastoma tumor cells. Culture medium with or without neutrophils was added to the bottom of the well to see whether transmigration of tumor cells was promoted by neutrophils (Figure 4A). Tumor cells were allowed to migrate for 24 h after which the transwell was removed and an image was made of the bottom of the culture well to quantify the number of transmigrated tumor cells. In accordance with the wound-healing assay, the presence of neutrophils on the bottom of the culture well significantly increased transmigration of BTIC10 and BTIC13 glioblastoma tumor cells (Figure 4B). As there was no direct contact between neutrophils and tumor cells in this assay, this result indicates that soluble factor(s) were involved in the promotion of tumor cell migration by neutrophils.

## 4. Discussion

In this study, we investigated the role of neutrophils in affecting migration of glioblastoma tumor cells after biopsy. Understanding the underlying mechanism behind this side effect would possibly allow to employ supplementary therapeutic strategies during biopsy for its prevention.

In direct comparison to the study by Alieva et al. [9], we found that systemic neutrophil depletion by injection with Ly6G antibodies prevented biopsy-induced tumor cell migration. We observe a decrease in the percentage of neutrophils in the blood compared to controls, but not a complete absence, at day 5 after depletion. Indeed, previous studies point towards biological effects preventing complete depletion of neutrophils in mice [29,30,32]. Insufficient antibody presence could not explain the rebound of neutrophils as increased dose or application frequency did not affect this result [29,30,32].

Despite the limited effect of neutrophil depletion, we do report a clear reduction in the number of macrophages/microglia in the glioblastoma as an effect of Ly6G-antibody injection. Given the absence of Ly6G expression on both macrophages and monocytes [33], the reduction of macrophages in the tumor is unlikely to be a direct result of treatment with Ly6G antibodies. Indeed, Daley et al. provide evidence that Ly6G antibody treatment does not affect blood monocyte counts while blood neutrophils are depleted [33]. Instead, our data indicate that neutrophils are involved in the recruitment of macrophages to the tumor.

Neutrophils have been shown to contribute to monocyte recruitment to sites of inflammation [33,34]. Indeed, impaired monocyte recruitment has been seen in several acute and chronic models of inflammation where neutrophils were depleted with Ly6G or Gr-1 antibodies [33,34]. Several mechanisms have been proposed to mediate this effect, including neutrophil release of CCR2 ligands (CCL-2, MIP-1α), leading to the recruitment of inflammatory monocytes expressing CCR2 [34]. We previously reported that the CCL-2/CCR2 axis was involved in the biopsy-induced recruitment of inflammatory monocytes to GL261 glioblastoma as CCL-2 antibodies could be used to block their recruitment [9]. It could be speculated that the same mechanism mediates neutrophil-dependent monocyte recruitment to the tumor. Indeed, it has previously been demonstrated in hepatocellular carcinoma that tumor-associated neutrophils recruited macrophages via CCL-2 [35].

To summarize, our data show that neutrophils are involved in promoting in vivo glioma tumor cell migration after biopsy, and this may in part be achieved through recruitment of blood monocytes to the tumor.

Next, we showed that neutrophils promote the in vitro migration of human glioblastoma tumor cell lines, indicating an additional direct role for neutrophils in affecting glioblastoma tumor-cell migration. Employing two well-established migration assays, the transwell migration assay and wound-healing assay, we observed increased transmigration and faster wound-closure respectively of both BTIC10 and BTIC13 glioblastoma tumor cell lines in the presence of neutrophils [36]. The increased transmigration of glioblastoma tumor cells by neutrophils implies that soluble factor(s) are responsible for the migratory effect. These results provide some explanation to clinical observations in glioblastoma patients, where high numbers of blood and tumor-infiltrating neutrophils are associated with higher glioma grade, poor prognosis, and resistance to therapy [21,37,38]. Further studies should be directed toward correlating results from animal models to results in humans and also toward identifying the soluble factors behind this direct effect of neutrophils on tumor cell migration.

In summary, we provide evidence for an indirect and direct involvement of neutrophils in the pro-migratory effect of glioblastoma tumor cells after biopsy. This finding warrants additional research that may provide, at its late end, a rationale for the development of therapeutic strategies targeting neutrophils and/or the soluble factor(s) derived from neutrophils responsible for increased tumor migration, in order to reduce the risk of developing tumor invasion following biopsy or tumor resection.

## Figures and Tables

**Figure 1 cells-11-02196-f001:**
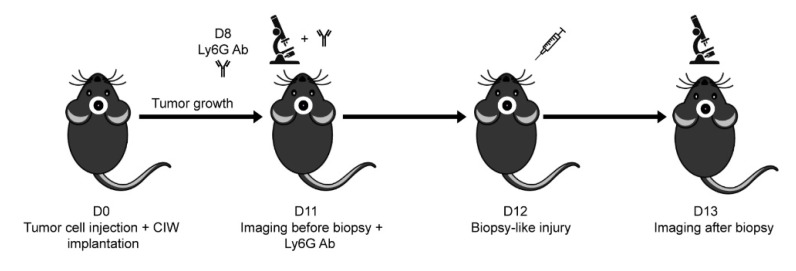
Cartoon showing set-up of experiment to study the effect of biopsy on in vivo glioma tumor cell migration in mice depleted of neutrophils with Ly6G antibodies.

**Figure 2 cells-11-02196-f002:**
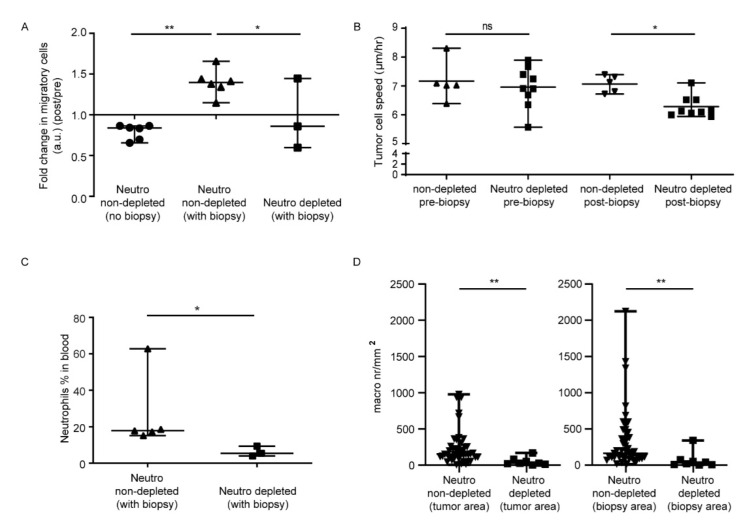
Biopsy induces a neutrophil-dependent increase in the in vivo motility of mouse glioma tumor cells. (**A**) Direct comparison of the fold changes (post day 13/pre day 11) in percentage of migratory glioma tumor cells before/after biopsy in 3 neutrophil-depleted biopsied mice to that of six control mice with and six control mice without biopsy from Alieva et al. [9]. Tracks from multiple imaging positions within one mouse were pooled; * *p* < 0.05, ** *p* < 0.01 in one way ANOVA. Bars represent median values with range. (**B**) Speed of migratory tumor cells of neutrophil depleted mice (before and after biopsy) compared to that of six control mice from Alieva et al. [9]. Separate imaging positions are shown. (**C**) Neutrophil percentage in the blood of neutrophil-depleted mice compared to that of control mice from Alieva et al. [9]. (**D**) Enumeration of the numbers of (F4/80^+^) macrophages/microglia by immunostaining of tumor tissue cryosections in biopsied mice without depletion from Alieva et al. [9] versus neutrophil-depleted mice. Areas with beads are labeled as biopsy area and random non-biopsied tumor tissue as tumor area.

**Figure 3 cells-11-02196-f003:**
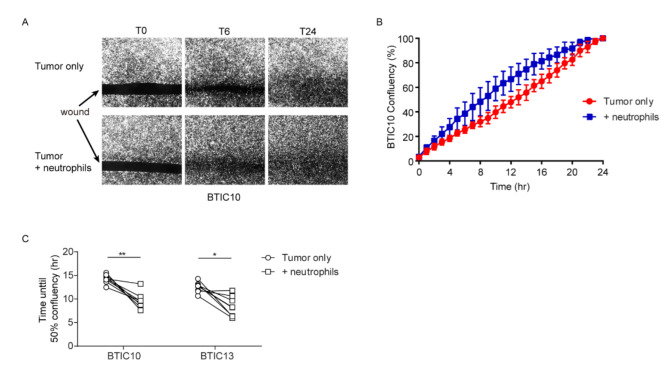
Neutrophils promote in vitro wound-closure of human glioblastoma tumor cells. (**A**) Representative images of BTIC13 at 0, 6, and 24 h after start of incubation in the wound-healing assay with or without neutrophils. Comparison of wound closure at 6 h shows increased migration of glioblastoma cells in the presence of neutrophils. (**B**) Representative graph of image analysis of BTIC13 showing the median confluency of the cell-free gap with range over time in the presence and absence of neutrophils for all donors. Confluency at 24 h was defined as 100%. (**C**) The analysis method depicted in A and B was used to determine the time in which 50% confluency was reached in the wound-healing assay for the different glioblastoma (BTIC10 and BTIC13) tumor cell lines. Dots represent different neutrophil donors. * *p* < 0.05, ** *p* < 0.01 in two-tailed Wilcoxon matched-pairs signed rank test.

**Figure 4 cells-11-02196-f004:**
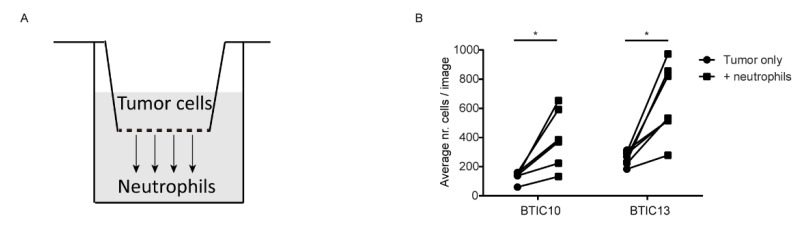
Soluble factor(s) from neutrophils increase transmigration of human glioblastoma tumor cells. (**A**) Cartoon showing the set-up of the transwell-migration assay. Tumor cells are seeded on top of the transwell with neutrophils on the bottom of the well. (**B**) Graph comparing average nr. cells/image (of 5 images) for glioblastoma (BTIC10 and BTIC13) tumor cell lines transmigrated in the presence or absence of neutrophils seeded on the bottom of the well. Each dot represents a different neutrophil donor. * *p* < 0.05, in two-tailed Wilcoxon matched-pairs signed rank test.

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
