# Peer review of "Neutrophils Promote Glioblastoma Tumor Cell Migration after Biopsy"

_cells, 2022, doi:10.3390/cells11142196_

Round 1

Reviewer 1 Report

This manuscript presents experimental evidence of the involvement of neutrophils in the pro-migratory effect of glioblastoma tumor cells after biopsy or biopsy-like injury. No major concerns are raised. There are a few minor issues:

1. In the Tumor cell lines (under Materials & Methods), only the human BTIC cell lines were listed. The results presented in Figure 1 were mainly performed using GL261 mouse glioma cells expressing a nuclear fluorescent protein (what is this protein? Is it GFP?), but this cell line was not given in Materials & Methods. It would be nice to list all the cell lines used in this study under the Tumor cell lines. 

2. In Figure 1B, the results from the Neutro-depleted (no biopsy) group are missing. This group can be used for the comparison with the group of Neutro non-depleted (no biopsy).  It seems inappropriate to compare the number from the group of Neutro non-depleted (no biopsy, on D11) to the number from the group of Neutro non-depleted (with biopsy, on D13) because there are two changing parameters for the two comparing groups: biopsy vs non-biopsy, and the imaging performed on D11 vs on D13 (tumor sizes and tumor cell numbers and activities on D13 may be quite different from those on D11). When the results of Neutro-depleted (no biopsy) are shown in Figure 1B, the text on Page 5 (and perhaps the legend of Figure 1) may also need to be changed or modified accordingly. 

3. Line 221 to Line 225 on Page 5, the meaning of the phrase "non-depleted tumors" or "non-depleted controls" is unclear to the readers. Do authors mean "the controls without neutrophil depletion"?

4. The format of the names of journals is not consistent throughout in the reference list. Some are capitalized for all journal names. but some others are not.

Author Response

Response

We want to thank the reviewer for taking the time to read our manuscript and the valuable comments. Our point-by-point responses can be find below.

  1. We have added the GL261 mouse glioma cell line to the Tumor cell lines section at line 93.
  2. Sorry for the confusion. We agree with the reviewer that it is inappropriate to compare D11 no biopsy to D13 with biopsy, but this is not what we do in Figure 1B. We calculate the fold change of D13 over D11 for each condition. In the no biopsy condition we calculate the fold change of D13 over D11 in case we do not perform a biopsy. Indeed, as hypothesized by the reviewer, the tumor size increase at D13 inducing a reduction in migrated cells. Hence the fold change <1 for this condition. In contrast, biopsy increases the migrated cells to a fold change >1. Neutrophil depletion prevents this increase in migrated cells. We have adjusted the text (lines 233, 239, 240) and the figure legend (line 253) to clarify.
  3. Indeed we mean "the controls without neutrophil depletion". We have adjusted this in the text.
  4. We have made the reference list consistent.

Reviewer 2 Report

In the manuscript of Na Chen et al. the role of neutrophils in migration of glioblastoma tumor cells is being investigated. It has been hypothesized that neutrophils are involved in tumor invasion and expansion after biopsy.

The presented results are meaningful and scientifically sound.

However, I have some comments:

1.    Arrange the references in the manuscript in order.

2.    Materials and methods Section «Mice» - indicate the number of animals in groups and make a reference to figure 1 (experimental scheme).

3.    Materials and methods Section «Human blood and neutrophil isolation» - indicate the number of blood donors.

Author Response

Response

We want to thank the reviewer for taking the time to read our manuscript and the valuable comments. Our point-by-point responses can be find below.

  1. We have arranged the references.
  2. We have indicated the number of mice in the Materials and Methods section Mice (line 118, 123, 125 and 126) and have made a reference to Figure 1A (line 130).
  3. We have indicated the number of blood donors in the Materials and Methods section Wound-healing Assay (line 180) and Transwell migration assay (line198).

Reviewer 3 Report

The second author of the current manuscript previously reported that biopsy-like injury induced in an orthotopic murine model of brain tumor promotes tumor cell growth and migration and that recruitment of macrophages through the CCR2-CCL2 axis is important for the tumor progression induced by biopsy. In the present manuscript, the authors aimed to investigate the contribution of neutrophils in biopsy-associated enhanced tumor cell migration using a similar orthotopic mouse glial model and intravital imaging technique. With the depletion of neutrophils using Ly6G antibody, the authors found reduced fold change in migratory cancer cells post- and pre-injury induction. However, the authors noted that the neutrophil depletion technique led to a decrease in the number of macrophages in the tumor. In order to determine the direct effect of neutrophils in promoting cancer cell migration, the authors next took the help of in vitro cell migration assays. The authors found a faster wound healing when cancer cell lines were co-cultured with neutrophils. Similarly, a greater number of cancer cells migrated to the transwell bottom chamber seeded with neutrophils compared to media control. Overall, the findings indicate a role of neutrophils in promoting the migration of glioblastoma cell lines possibly through the secretion of diffusible factors.

The novelty of the current study lies in that it is looking into the possibility of neutrophils promoting cancer cell migration in the context of biopsy-like injuries. However, the study does not present a comprehensive investigation of neutrophils’ role post biopsy-like injuries in the in vivo tumor model. In vitro data showing neutrophils directly promoting cancer cell migration are interesting and in agreement with previous findings in other cancer type (https://doi.org/10.3892/or.2017.5942). The mechanistic aspect of how cancer cells acquire better migration ability in the presence of neutrophils or neutrophil-derived factor/s is, however, missing. Addressing the following points will make the study more focused and complete.

Major concerns:

In vivo system

1.     Ly6G antibody depletion experiments should have been accompanied by IP injection of control animals with an equivalent amount of isotype-matched antibody (1A8).

2.     The authors should investigate if neutrophils are indeed involved in monocyte/macrophage recruitment in the brain tumor model or it is some non-specific effect of antibody-mediated depletion technique by incorporating alternative ways such as treating with pepducin (https://doi.org/10.1172/JCI61067).

3.     A kinetic study on the presence of neutrophils and macrophages during tumor development will provide a better understanding of how neutrophils may impact monocyte recruitment to tumors. Neutrophils post-injury should be harvested and tested for the expression of potential monocyte recruiting chemokines, especially CCL family members (as mentioned in the discussion).

In vitro system

1.     The authors should determine whether neutrophils promote glioblastoma cell line proliferation. The use of mitomycin c during the wound healing assay will help to tease apart the effect of proliferation vs. cell migration.

2.     The transwell assay results indicate that some diffusible factors generated and secreted in the system are inducing cancer cell migration. The authors must address the mechanistic aspect of this phenotype. Are the diffusible factors simply chemoattractants for cancer cells or do the factors induce epithelial-to-mesenchymal transition in cancer cells, a well-established transition process involved in increased cancer cell migration and invasion?

Minor comment:

a.     Cancer cells generally migrate way slower than immune cells such as neutrophils. Taking live imaging for 2 hrs with 20 min intervals would generate only 6-time frames, which seems too short to provide meaningful information. The authors should provide some rationale for selecting the parameters for the timescale.

b.     Line 290 and 301: “A soluble factor from neutrophils increases transmigration of human glioblastoma cells”

“this result indicates that a soluble factor 301 was involved in the promotion of tumor cell migration by neutrophils”

Multiple factors can be involved

Author Response

Response

We want to thank the reviewer for his or her time to read our manuscript and for the elaborate and valuable comments. Our point-by-point responses can be find below.

Major concerns in vivo system

  1. This is a valid point. We have previously reported (Alieva et al 2017) that a control IgG did not change GL261 migration nor cell division. We found it ethically undesirable to repeat this control. We have now added the reference to this control in the manuscript (line 246).
  2. CCL-2 is a chemokine secreted by tumor and stromal cells and mediates recruitment of monocytes and neutrophils, both expressing the receptor CXCR2. The approach suggested by the reviewer, Pepducin, inhibits CXCR1/2, so would impact on the recruitment of both, neutrophils and monocytes. As an alternative approach for pepducin we have previously shown (Alieva et al 2017) that anti-CCL2 prevented macrophage recruitment. So indeed CXCR1/2 seems to play a role but it is not a non-specific effect. We have clarified this in the manuscript (line 363 and 365). Additionally we have included a reference in another tumor type that suggests that neutrophils participate in macrophages recruitment in tumor settings (line 367).
  3. These are great suggestions for future studies but are beyond the scope of this manuscript.

Major concerns in vitro system

  1. We agree with the reviewer that this is an interesting and important point. We chose a different approach than suggested by the reviewer to more directly check the effect of neutrophils on proliferation. We seeded green BTIC at a low confluency with or without neutrophils and checked the fold change in green signal at 22 hours compared to the first timepoint as a measure for proliferation. Interestingly proliferation was significantly lower with neutrophils than without when comparing 8 to 9 donors in 2 different experiments. Thus, increased proliferation of tumor cells in the presence of neutrophils could not be the explanation for the faster wound healing or the increased transwell migration. We have now added this data in the text of the manuscript (line 298-300) and as Supplementary Fig 1.
  2. As glioblastoma cells are not of epithelial origin EMT is not applicable to this type of tumor.

Minor comments

  1. We did not visualize neutrophils in the intravital images but GL261 tumor cells. The goal was to visualize GL261 tumor cell motility and for this cell type the 20 min intervals are appropriate, as we identified in Alieva et al 2017.
  2. We agree, we have changed the text accordingly (line 31, 315, 326, 329 379, 389).

Reviewer 4 Report

In this study, Chen et al. investigated the role of neutrophils in affecting migration of glioblastoma tumor cells after biopsy. The authors showed that neutrophils depletion by Ly6G antibodies prevented in vivo glioma tumor cell migration after biopsy. This may be achieved through recruitment of blood monocytes to the tumor by neutrophils and soluble factor secreted by neutrophils. The manuscript is well-written and straight forward. However, the strength of the results is limited by the small number of animals used in some experiments and the lack of clarity on the number of independent repetitions performed. Some specific points are as follows:

1. Fig. 1B and 1D, only 3 mice were used in neutrophils depleted group. The number of mice per group should be increased to at least n=6 cumulative of two independent experiments at least.

2. The authors must provide the proof that the in-vivo biopsy caused inflammation in tumor.

3. The authors need to check that immune cell profiles in spleen and tumor in mice after biopsy.

Author Response

Response

We want to thank the reviewer for his or her time to read our manuscript and for the valuable comments. We have clarified the number of repetitions in the manuscript. Our point-by-point responses can be find below.

  1. We feel the results are sufficiently clear and in fact already significant with three mice. We also want to emphasize that three positions per mouse at each imaging session were measured leading to 1081-3562 measured tracks per mouse. Additionally given the fact that intravital imaging allows us to perform repetitive imaging in the same mouse, where we can compare the impact of neutrophil depletion on the baseline migration in each single mouse, strengthens our results and withdraws the necessity of bigger animal groups used in classical experiments. It would thus be unethical to increase the animal size. As indicated by the reviewer the experiments were indeed performed in two independent groups. We clarified this in the text (line 123-126).
  2. This is a valid point which we already demonstrated previously. In Alieva et 2017 we observed a 3.3 fold increase in the number of Gr1+neutrophils and a 2 fold increase in the number of F4/80+ macrophages/microglia at the biopsy site. We have clarified this in the manuscript at line 223.
  3. We do not understand the relevance of spleen immune cell profiles for this project. We agree that the immune cell profile of the tumor is highly relevant. Indeed we looked at the number of macrophages in the tumors in Figure 1E. Since remaining neutrophils might be saturated with Ly6G depletion mAbs, we could unfortunately not use this antibody nor the Gr-1 antibody directed to Ly6C and Ly6G to detect neutrophils in tumor tissue.

Round 2

Reviewer 1 Report

The authors revised the manuscript properly. There is no further concerns from this reviewer.

Author Response

This is great news. Thanks for your time and effort!

Reviewer 2 Report

The authors have made all the required changes. The manuscript can be accepted for publication.

Author Response

(The authors gave the same response as above.)

Reviewer 3 Report

"CCL-2 is a chemokine secreted by tumor and stromal cells and mediates recruitment of monocytes and neutrophils, both expressing the receptor CXCR2"

The authors should clearly explain what role CXCR2 is playing in this context. The following sentence sounds like CCL2 is mediating monocyte and neutrophil recruitment through CXCR2. However, the receptor for CCL2 is CCR2. It might be a typo.

Author Response

Dear reviewer,

Thanks for pointing out this typo. We have changed CXCR2 to CCR2 at line 85 in the revised manuscript. We are thankful for the thorough review and pleased we have addressed all comments raised in round 1.

Reviewer 4 Report

The authors have generated a revised version of the manuscript in which the major concerns have been addressed.

Author Response

(The authors gave the same response as above.)
